

# Development and characterization of 24 polymorphic microsatellite *loci* for the freshwater fish *Ichthyoelephas longirostris* (Characiformes: Prochilodontidae)

Ricardo M. Landínez-García and  Edna J. Márquez

Facultad de Ciencias, Universidad Nacional de Colombia, Medellin, Antioquia, Colombia

## ABSTRACT

The Neotropical freshwater fish *Ichthyoelephas longirostris* (Characiformes: prochilodontidae) is a short-distance migratory species endemic to Colombia. This study developed for the first time a set of 24 polymorphic microsatellite *loci* by using next-generation sequencing to explore the population genetics of this commercially exploited species. Nineteen of these *loci* were used to assess the genetic diversity and structure of 193 *I. longirostris* in three Colombian rivers of the Magdalena basin. Results showed that a single genetic stock circulates in the Cauca River, whereas other single different genetic stock is present in the rivers Samaná Norte and San Bartolomé-Magdalena. Additionally, *I. longirostris* was genetically different among and across rivers. This first insight about the population genetic structure of *I. longirostris* is crucial for monitoring the genetic diversity, the management and conservation of its populations, and complement the genetic studies in Prochilodontidae.

## INTRODUCTION

*Ichthyoelephas longirostris* (Steindachner, 1879) is a Colombian endemic fish of importance in the commercial and subsistence fisheries. This freshwater fish is a member of the family Prochilodontidae, which comprises three genera (*Ichthyoelephas*, *Prochilodus* and *Semaprochilodus*) and 21 species that occur in the major river basins of South America (*Vari, 1983*). In the Colombian red list of threatened freshwater fishes, *I. longirostris* is considered an endangered species based on some criteria that include scarce biological and ecological information, restricted distribution (tributaries of Magdalena and Ranchería basins), infrequent catches, disappearance in some floodplain lakes and rivers (Ranchería) and the habitat degradation by anthropogenic activities (*Mojica et al., 2012*).

Furthermore, this detritivourous species prefers turbulent and clear waters (*Román-Valencia, 1993*) and has a short-distance migration range (approx. 20 km; *López-Casas et al., 2016*) suggesting that its populations may be genetically structured. Such behavior seems a generalized tendency within Prochilodontidae because other members of this

Corresponding author
Edna J. Márquez,
ejmarque@unal.edu.co,
ejmarque@gmail.com

family also have genetic stocks that coexist and co-migrate in the same hydrographic basin, such as *P. reticulatus* (*López-Macias et al., 2009*), *P. marggravii* (*Hatanaka & Galetti Jr, 2003*), *P. argenteus* (*Hatanaka, Henrique-Silva & Galetti Jr, 2006*; *Barroca et al., 2012a*), *P. costatus* (*Barroca et al., 2012b*) and *P. magdalenae* (*Orozco Berdugo & Narváz Barandica, 2014*). However, the population genetics of *I. longirostris* remain unknown, although this information is necessary to develop management and conservation policies for this species.

For population genetic studies in the Prochilodontidae family, polymorphic microsatellites have been developed in members of genera *Prochilodus* (*Barbosa et al., 2006*; *Barbosa et al., 2008*; *Carvalho-Costa, Hatanaka & Galetti, 2006*; *Yazbeck & Kalapothakis, 2007*; *Rueda et al., 2011*) and *Semaprochilodus* (*Passos et al., 2010*). No such markers have been developed to date for congeners of *I. longirostris* limiting our ability to resolve fine-scale differentiation and subdivision patterns among populations. Moreover, cross-amplification of microsatellite loci developed from other species often presents problems such as allele size homoplasy, unsuccessful amplification in phylogenetically distant species, lower levels of polymorphism, null alleles, broken repeat motifs and amplification of non-orthologous loci (*Primmer et al., 2005*; *Barbará et al., 2007*; *Rutkowski, Sielezniew & Szostak, 2009*; *Yue, Balazs & Laszlo, 2010*).

Therefore, it is highly recommended to develop species-specific molecular markers, which has been greatly facilitated by next generation sequencing technologies (*Castoe et al., 2010*; *Ekblom & Galindo, 2011*). In addition to other advantages, this approach permits the rapid selection of long repeat motifs to prevent the genotyping problems associated with di-nucleotide (*Gardner et al., 2011*; *Fernandez-Silva et al., 2013*; *Schoebel et al., 2013*). Thus, this study developed *de novo* molecular markers for future population genetics studies of *I. longirostris* using next-generation DNA sequencing and bioinformatics. In addition, these novel microsatellite loci were used to assess whether *I. longirostris* comprise genetically differentiated populations in sections of three Colombian rivers of the Magdalena basin.

## MATERIALS & METHODS

This study analyzed a total of 193 preserved tissues of *I. longirostris* from three rivers in the Magdalena-Cauca basin that will be influenced by two hydropower station projects: Cauca River (Ituango project), and the Samaná Norte and San Bartolomé rivers (Porvenir II project). All these samples were provided by Integral S.A., through two scientific cooperation agreements (19th September 2013; Grant CT-2013-002443). In the Cauca River, the samples came from three out of eight sections sampled (Fig. 1A) because the number of individuals was extremely low in the other sections: S1 and S2/3 are river sites in the department of Antioquia, whereas S6 comprise two floodplain lakes in the department of Bolívar. Sections S1 and S2/3 are respectively upstream and downstream of an area that exhibits steep topography, rapids, geomorphologic peculiarities, riverbed narrowing, and drastic changes in the water velocities and the slopes. Similarly, the rivers Samaná Norte and San Bartolomé-Magdalena (Fig. 1B) exhibit canyons associated with rough topography and pronounced changes in the slope. These characteristics of the three rivers may limit the migration of several fishes.

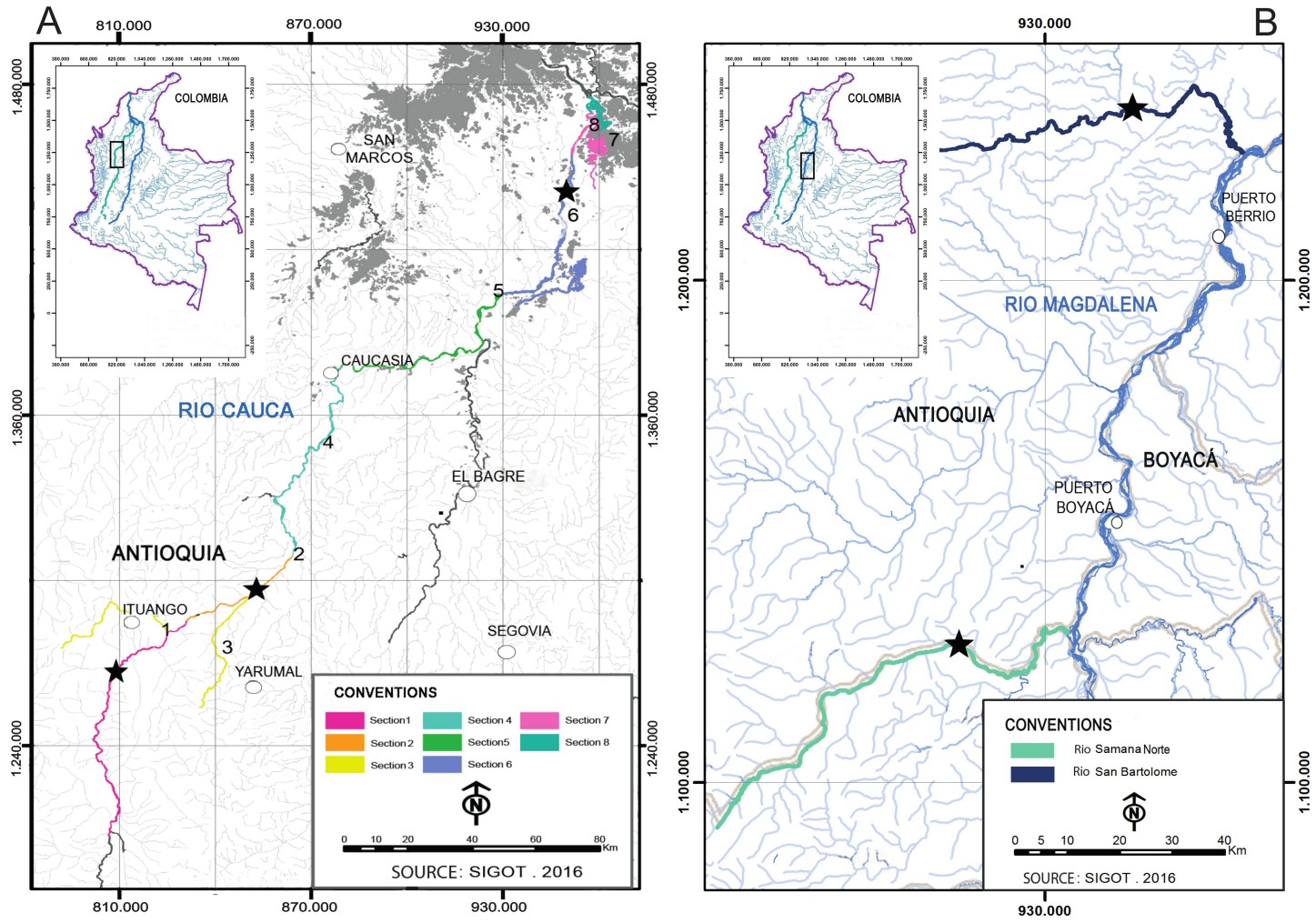

**Figure 1** Sampling sites (stars) of *I. longirostris* in the Colombian rivers Cauca (A), San Bartolomé and Samaná Norte (B).

DNA isolations were performed with the commercial kit GeneJET DNA purification (Thermo Scientific) following the manufacturer's instructions. To identify microsatellite regions and develop primers to amplify them, low-coverage genome sequencing was carried out with the Illumina MiSeq v2 instrument using the Nextera library preparation kits. This sequencing process generated paired-end reads of 250 bases that were cleaned using Prinseq-lite v0.20.4 (*Schmieder & Edwards, 2011*) to eliminate low quality regions at both ends and remove reads that were duplicated or <50 bases in length. The genome assembly of reads was performed with Abyss v1.3.5 (*Simpson et al., 2009*) using a kmer 64 and the contigs were analyzed with the PAL_FINDER v.0.02.03 software (*Castoe et al., 2010*) to extract those that contained perfect tri-, tetra- and pentanucleotide microsatellites. The primer-pairs for microsatellite loci amplification were designed from their flanking sequences by using the Primer3 v.2.0 software (*Rozen & Skaletsky, 2000*). Additionally, the potential amplifiable loci were submitted to electronic PCRs (*Rotmistrovsky, Jang & Schuler, 2004*) for verifying *in silico* the correct primer alignment (http://www.ncbi.nlm.nih.gov/tools/epcr/).

A total of 40 microsatellites were selected for optimization and polymorphism analysis in *I. longirostris.* Preliminary tests of standard PCR conditions (*Sambrook, Fritsch & Maniatis, 2001*) were carried out in 15 DNA samples and the amplicons were separated in 10% polyacrylamide gel in a Mini Protean® Tetra vertical electrophoresis cell (Biorad™) run at 100 volts for 45 min and visualized by silver-stain. Polymorphic loci were selected based on criteria of amplification in all samples, band resolution, specificity, size (from 100 to 400 bp) and ability to detect heterozygotes in the different samples analyzed. Then, a set of 24 polymorphic microsatellite loci that met these criteria and amplified consistently were selected and fluorescently labelled for further genotyping of 28 samples of *I. longirostris.* Finally, a subset of 19 loci were selected to evaluate the genetic diversity and genetic structure in 193 samples from three Colombian rivers.

PCR reactions in volumes of 10 µl containing final concentrations of 1×buffer (Invitrogen), 2–4 ng/µl of template DNA, 2.5% formamide (Sigma) 0.35 pmoles/µl labelled forward primer (either FAM6, VIC, NED or PET, Applied Biosystems), 0.5 pmoles/µl reverse primer (Macrogen), 0.2 mM dNTPs (Thermo Scientific), 0.05 U/µl Platinum™ Taq DNA Polymerase (Invitrogen) and 2.5 mM $MgCl_2$. The PCR amplifications were performed on a thermocycler T100 (BioRad) with an initial denaturation step of 95 °C for 3 min, followed by 32 cycles consisting of a denaturation step of 90 °C for 22 s and an annealing step of 57 °C for 16 s. The extension step and a final elongation were absent in this thermal profile. Finally, the PCR products were submitted to electrophoresis on an automated sequencer ABI 3730 XL (Applied Biosystems) using LIZ500 (Applied Biosystems) as internal molecular size. Allelic fragments were denoted according their molecular size and scored using GeneMapper 4.0 (Applied Biosystems).

Tests for departures of Hardy–Weinberg and linkage equilibria and the estimation of the observed ($H_O$) and expected ($H_E$) heterozygosities were performed using Arlequin v.3.5.2.2 (*Excoffier, Laval & Schneider, 2005*). Statistical significance in multiple comparison was adjusted applying the sequential Bonferroni correction (*Rice, 1989*). The software GenAlEx v.6.501 (*Peakall & Smouse, 2006*) was used to estimate the average number of alleles per locus. Potential genotyping errors were evaluated by using Micro-Checker v.2.2.3 software (*Van Oosterhout et al., 2004*). The polymorphism information content (PIC) for each marker was determined using the program PICcalc (*Nagy et al., 2012*).

Genetic diversity was estimated by calculating the average number of alleles per locus, observed and expected average heterozygosities and fixation index. Non-geographical genetic differentiation among samples was tested the Bayesian analysis of population partitioning using Structure v.2.3.4 (*Pritchard, Stephens & Donnelly, 2000*). This analysis was performed with 100,000 Monte Carlo Markov Chain (MCMC) steps and 10,000 iterations as burn-in, run length that reached the convergence. Parameters included admixture model, correlated frequencies and the LOCPRIOR option for improving the performance of the algorithm when the signal of the structure is relatively weak (*Hubisz et al., 2009*). Each analysis was repeated 20 times for each simulated *K* value, which ranged from 1 to 8 groups. Then, the best estimate of genetic stocks (*K*) was calculated using Δ*K ad hoc* statistic (*Evanno, Regnaut & Goudet, 2005*) with STRUCTURE Harvester (*Earl & VonHoldt, 2012*). Results of independent STRUCTURE runs were summarized using

CLUMPP v.1.1.2b (*Jakobsson & Rosenberg, 2007*), setting the parameters to their default values and the algorithm full search with the function G'normalized to guarantee we would find the optimal alignment of clusters across multiple runs. The Q-matrix obtained was plotted in a histogram displaying the ancestry of each individual in each population using DISTRUCT v.1.1 (*Rosenberg, 2004*). In addition, genetic differentiation among geographical samples was calculated by the standardized statistics $F'_{ST}$ (*Meirmans, 2006*) and Jost's *D*est (*Meirmans & Hedrick, 2011*) and analysis of molecular variance, AMOVA (*Meirmans, 2006*) included in GenAlex v.6.502 (*Peakall & Smouse, 2006*). Furthermore, the diploid genotypes of 19 loci (38 variables) in 193 individuals were submitted to Discriminant analysis of Principal Components to examine other grouping of the samples using RWizard (http://www.ipez.es/RWizard/).

## RESULTS

All 24 polymorphic microsatellite loci initially selected for additional testing (Table 1) showed clearly defined peaks and the absence of stutter bands in the chromatograms. The number of alleles per locus ranged from 4 to 18, with an average number of 8.5 alleles/locus and average observed heterozygosity ($H_o$) of 0.669. Additionally, allelic frequencies of 20 loci were concordant with Hardy-Weinberg and linkage equilibria after sequential Bonferroni correction and no evidence of null alleles or scoring errors were detected by Micro-checker. Moreover, the PIC values ranged from 0.375 to 0.871 (average: 0.733) indicating that these markers are highly informative (*Botstein et al., 1980*). Nineteen of these loci were subsequently used to explore the population genetics of *I. longirostris* in a greater sample, whereas the other five are awaiting a similar analysis.

In the rivers, 3 of the 19 loci (Ilo21, Ilo11, Ilo3) departed from Hardy-Weinberg equilibrium expectations in all samples evaluated (Table 2). The average number alleles per locus was higher in Samaná Norte (11.95) and Cauca S2/3 (11. 05) followed by Cauca S6 (9.16), Cauca S1 (8.84) and San Bartolomé-Magdalena (8.32). Additionally, the higher values of observed and expected heterozygosities were found in Cauca S2/3 ($H_o$: 0.767; $H_e$: 0.798) and Samaná Norte ($H_o$: 0.742; $H_e$: 0.796) followed by Cauca S1 ($H_o$: 0.713; $H_e$: 0.773), Cauca S6 ($H_o$: 0.701; $H_e$: 0.771) and San Bartolomé-Magdalena ($H_o$: 0.692; $H_e$: 0.768).

The three approaches for measuring genetic differences revealed contrasting results.The Bayesian analysis showed the presence of two genetic stocks ($\Delta K = 2$), one predominantly in the Cauca River and the other one in the rivers San Bartolomé and Samaná Norte (Fig. 2A). Although $K = 2$ was the most supported number of clusters using the $\Delta K$ method, an additional clustering pattern ($K = 4$) was examined to compare it with the other approaches (Fig. 2B). This latter analysis showed the same tendency of clustering in two major stocks and two minor stocks with non-homogenous distribution (Fig. 2B). However, the discriminant analysis of principal components and AMOVA found significant genetic differences of *I. longirostris* among (Figs. 2C and 2E; $F_{ST(0.001)} = 0.010$; $P = 0.000$) rivers. In addition, within the Cauca River, the discriminant analysis of principal components (Fig. 2D) displayed differences among the three sections examined, whereas AMOVA

**Table 1  Primer sequences and characteristics of 24 polymorphic microsatellite *loci* identified in *Ichthyoelephas longirostris*.**

| Name *locus* | Primer sequence for forward (F) and reverse (R) (5′ − 3′) | Repeat motif | Number of alelles | $H_O$ | $H_E$ | PIC | $P$ |
|---|---|---|---|---|---|---|---|
| Ilo01 | F: TGCATCTGAGCTGATGGAGG<br>R: AGTCTCTCTGCAGGTTGGGG | $(AAAAC)_n$ | 8 | 0.857 | 0.828 | 0.789 | 0.326 |
| Ilo03 | F: CAGATGCAGCTGAACACGG<br>R: TTGTAAACTGGCAGTGTGTTAAACC | $(AAAC)_n$ | 5 | 0.571 | 0.509 | 0.411 | 0.968 |
| Ilo04 | F: GAAGCTGGCGAATAGAAGGC<br>R: TGACCTACTGTGAAACTGGGG | $(AAAC)_n$ | 7 | 0.526 | 0.605 | 0.555 | 0.125 |
| Ilo05 | F: GAAGGAACTGAGGTGCAGGG<br>R: CACATCTCCCTCTGTATCCCC | $(AAAG)_n$ | 9 | 0.773 | 0.791 | 0.773 | 0.886 |
| Ilo06 | F: TCCGTTGATGTAACAACATTAGCC<br>R: GCTCCCTGTGCTCTTCTGC | $(AAAG)_n$ | 10 | 0.741 | 0.848 | 0.832 | 0.420 |
| Ilo08 | F: GGTTGGGAGTGCCAGATAGG<br>R: AGTGCAGTGCTCAGTCCAGC | $(AAAG)_n$ | 8 | 0.679 | 0.742 | 0.702 | 0.218 |
| Ilo09 | F: ATGTTTGTGGCATCACCAGG<br>R: CTGGCAGTGCTACCTCAACC | $(AAATC)_n$ | 9 | 0.821 | 0.799 | 0.755 | **0.022** |
| Ilo10 | F: TACGACAGCTGACTGACCCG<br>R: CCCCTAAGAGACAACCGACC | $(AAC)_n$ | 8 | 0.714 | 0.808 | 0.763 | 0.693 |
| Ilo11 | F: TGTCGTGTCATGTTGTGTCG<br>R: CCCTGTACATGTCCTTCAGAGC | $(AACAT)_n$ | 5 | 0.308 | 0.609 | 0.548 | **0.002** |
| Ilo12 | F: TTGGACCAGATGTGTTTGCC<br>R: TCCTCAGGCATCCTACTGCC | $(AACG)_n$ | 4 | 0.571 | 0.689 | 0.677 | 0.455 |
| Ilo15 | F: CATAGTAGTGTCATACAACACCTGTGC<br>R: TCATTAACCCGTTTGGTGAGG | $(AATG)_n$ | 8 | 0.714 | 0.838 | 0.799 | **0.013** |
| Ilo16 | F: AGTGTGCGGGGTTAAACTGC<br>R: CCTGCGGTAGACTGGTAATCC | $(AATG)_n$ | 8 | 0.630 | 0.661 | 0.602 | 0.861 |
| Ilo17 | F: GCAGATGCTTTGGAGTTCCC<br>R: TGGCATGATTATCAATGGGC | $(AATG)_n$ | 10 | 0.857 | 0.866 | 0.835 | 0.051 |
| Ilo18 | F: ATAACTCTGCACTTCGGGGC<br>R: ATCTAAACCGCATGTGAGCC | $(AATG)_n$ | 5 | 0.393 | 0.401 | 0.375 | 0.561 |
| Ilo20 | F: ATTTTCACTCGTCGAAGCCC<br>R: TGATGTAAACCACAGGCACG | $(AGGCT)_n$ | 8 | 0.714 | 0.762 | 0.749 | 0.210 |
| Ilo21 | F: TCCATAACTTGTTTTGCTGCG<br>R: AATCTATAGTCTGAGAGCAACGGC | $(AGT)_n$ | 18 | 0.75 | 0.886 | 0.871 | 0.232 |
| Ilo22 | F: AAAACAATGCGCTGAATGC<br>R: ATGTGTACGTGTATATATGCTGGC | $(ATAC)_n$ | 4 | 0.536 | 0.647 | 0.636 | 0.227 |
| Ilo23 | F: CCAAACTGCTCATTCTGGAGG<br>R: TGGGACGCTTCTTTAGCTCC | $(ATAC)_n$ | 10 | 0.857 | 0.881 | 0.865 | 0.680 |
| Ilo24 | F: ACTGCACACTTGAGATCTGGG<br>R: GGTACGTTAGCCAAACAGACTGG | $(ATCT)_n$ | 10 | 0.75 | 0.86 | 0.844 | 0.311 |
| Ilo26 | F: TTAAGAGCTCAGAGCGTGCG<br>R: TGTTTAGCAACTTATTTATGACCTATGACC | $(ATCT)_n$ | 11 | 0.815 | 0.853 | 0.837 | 0.137 |

**Table 1** (*continued*)

| Name *locus* | Primer sequence for forward (F) and reverse (R) (5′ − 3′) | Repeat motif | Number of alelles | $H_O$ | $H_E$ | PIC | $P$ |
|---|---|---|---|---|---|---|---|
| Ilo29 | F: ATCTATCTGACAGACTATCTGTTTATTCC<br>R: GAAGCACTCAGAGACAGACAGG | $(ATCT)_n$ | 8 | 0.667 | 0.861 | 0.845 | **0.071** |
| Ilo35 | F: GGATACCCTAAATTTCCTTTGGG<br>R: GCATCACAGCGTCAAGAACC | $(TCCG)_n$ | 11 | 0.250 | 0.935 | 0.871 | **0.000** |
| Ilo37 | F: CACACAAACACTCATCTTAAAAGTCTCC<br>R: GACCTGCGGAAAGAGAATGG | $(TCTG)_n$ | 12 | 0.821 | 0.885 | 0.856 | 0.241 |
| Ilo40 | F: CAGAGTTTTGGCCGTGAGG<br>R: CAGGGAGGAGTAGTGTCGGG | $(TTC)_n$ | 8 | 0.750 | 0.833 | 0.795 | 0.218 |

**Notes.**

$H_O$ and $H_E$, observed and expected heterozygosity estimated from 28 individuals, respectively; PIC, polymorphic information content; $P$, statistical significance for tests of departure of Hardy–Weinberg equilibrium.

showed low but significant genetic differences among the S1 and the other sections of the Cauca River but not between S2/3 and S6 (Table 3).

## DISCUSSION

This study developed a set of 24 microsatellite loci for population genetic studies of the Colombian endemic fish *I. longirostris*. These loci are polymorphic, highly informative and 19 of them exhibited abilities to detect reliable levels of genetic diversity and structure in three Colombian rivers. Thus, these microsatellite loci are suitable for future studies of diversity and population genetics of *I. longirostris*. Remaining loci are awaiting to test their usefulness for population genetic analysis in a greater sample.

The mean number alleles per locus of *I. longirostris* is similar to that found in *P. argenteus* using microsatellite loci with pentanucleotide motifs (*Hatanaka, Henrique-Silva & Galetti Jr, 2006*). However, as expected, this value is lower than those found in other studies that include a greater selection of dinucleotide motifs (*Yazbeck & Kalapothakis, 2007*; *Passos et al., 2010*; *Orozco Berdugo & Narváz Barandica, 2014*; *Braga-Silva & Galetti Jr, 2016*). Additionally, the levels of observed and expected heterozygosities are similar to those found in *P. lineatus* and *Semaprochilodus insignis* (*Yazbeck & Kalapothakis, 2007*; *Passos et al., 2010*) and higher than those found in *P. costatus* (*Braga-Silva & Galetti Jr, 2016*), *P. argenteus* (*Hatanaka, Henrique-Silva & Galetti Jr, 2006*) and *P. magdalenae* (except expected heterozygosity; (*Orozco Berdugo & Narváz Barandica, 2014*). The levels of observed heterozygosity are also similar to the values of average heterozygosity per species across loci found in 13 freshwater fish species, using microsatellite loci ($0.54 \pm 0.25$; *Dewoody & Avise, 2000*).

Low incidence of *I. longirostris* in the fisheries has been interpreted as a signal of the decline in population density (*Mojica et al., 2012*). However, diversity levels found in this study might suggest that the low incidence in traditional fisheries may also result from the preference of this species for turbulent waters, which provide refuge in steep topography and treacherous rock riverbeds impeding its capture. Alternatively, since *I. longirostris* seems to make short displacements during dry stations (*López-Casas et al., 2016*), it might

Landinez-García and Márquez (2016), *PeerJ*, DOI 10.7717/peerj.2419

**Table 2** Genetic diversity per locus and across loci in *I. longirostris* from the Colombian rivers Cauca, San Bartolomé-Magdalena and Samaná Norte.

| Locus | Ra | Cauca River-S6 (N = 25) | | | | Cauca River-S2/3 (N = 42) | | | | Cauca-River-S1 (N = 33) | | | | San Bartolomé-Magdalena (N = 23) | | | | Samaná Norte River (N = 70) | | | |
|---|---|---|---|---|---|---|---|---|---|---|---|---|---|---|---|---|---|---|---|---|---|
| | | Na | $H_o$ | $H_e$ | P | Na | $H_o$ | $H_e$ | P | Na | $H_o$ | $H_e$ | P | Na | $H_o$ | $H_e$ | P | Na | $H_o$ | $H_e$ | P |
| Ilo10 | 186–237 | 12 | 0.880 | 0.869 | 0.346 | 12 | 0.975 | 0.871 | 0.518 | 13 | 0.879 | 0.877 | 0.269 | 12 | 0.826 | 0.850 | 0.529 | 16 | 0.857 | 0.877 | 0.134 |
| Ilo40 | 125–164 | 10 | 0.960 | 0.888 | 0.226 | 13 | 0.929 | 0.870 | 0.148 | 12 | 0.909 | 0.883 | 0.622 | 12 | 0.913 | 0.885 | 0.297 | 12 | 0.886 | 0.859 | 0.184 |
| Ilo09 | 256–316 | 9 | 0.760 | 0.766 | 0.562 | 10 | 0.786 | 0.795 | 0.443 | 8 | 0.788 | 0.775 | 0.759 | 6 | 0.739 | 0.758 | 0.148 | 12 | 0.743 | 0.807 | 0.419 |
| Ilo12 | 161–193 | 6 | 0.720 | 0.752 | 0.136 | 9 | 0.881 | 0.759 | 0.620 | 7 | 0.606 | 0.735 | 0.214 | 5 | 0.545 | 0.710 | 0.084 | 8 | 0.750 | 0.761 | 0.525 |
| Ilo20 | 163–203 | 7 | 0.840 | 0.784 | 0.848 | 8 | 0.786 | 0.818 | 0.301 | 7 | 0.788 | 0.811 | 0.750 | 8 | 0.652 | 0.788 | 0.151 | 8 | 0.843 | 0.786 | 0.547 |
| Ilo17 | 244–308 | 10 | 0.720 | 0.835 | 0.386 | 13 | 0.878 | 0.849 | 0.327 | 9 | 1.000 | 0.838 | 0.069 | 9 | 0.913 | 0.855 | 0.885 | 14 | 0.786 | 0.874 | **0.037** |
| Ilo37 | 96–148 | 7 | 0.960 | 0.827 | 0.085 | 9 | 0.929 | 0.861 | **0.016** | 9 | 0.818 | 0.846 | 0.872 | 9 | 0.818 | 0.859 | 0.861 | 12 | 0.857 | 0.847 | 0.139 |
| Ilo01 | 195–245 | 10 | 0.760 | 0.816 | 0.656 | 9 | 0.905 | 0.868 | 0.435 | 10 | 0.758 | 0.841 | **0.003** | 8 | 0.864 | 0.816 | 0.310 | 10 | 0.897 | 0.800 | 0.737 |
| Ilo18 | 237–281 | 9 | 0.520 | 0.624 | 0.370 | 10 | 0.667 | 0.648 | 0.136 | 6 | 0.545 | 0.642 | 0.332 | 6 | 0.455 | 0.399 | 1.000 | 9 | 0.522 | 0.646 | **0.001** |
| Ilo26 | 102–232 | 11 | 0.720 | 0.868 | 0.176 | 14 | 0.829 | 0.916 | **0.005** | 11 | 0.788 | 0.877 | **0.036** | 12 | 0.818 | 0.907 | 0.066 | 16 | 0.806 | 0.902 | 0.224 |
| Ilo22 | 214–298 | 13 | 0.737 | 0.925 | 0.073 | 15 | 0.667 | 0.904 | **0.001** | 13 | 0.879 | 0.878 | 0.112 | 11 | 0.913 | 0.887 | 0.404 | 14 | 0.714 | 0.886 | **0.036** |
| Ilo15 | 152–200 | 7 | 0.680 | 0.811 | 0.428 | 10 | 0.833 | 0.821 | 0.714 | 8 | 0.758 | 0.819 | **0.030** | 8 | 0.591 | 0.846 | **0.005** | 9 | 0.786 | 0.813 | 0.472 |
| Ilo04 | 254–286 | 5 | 0.760 | 0.682 | 0.683 | 6 | 0.825 | 0.677 | **0.027** | 6 | 0.818 | 0.690 | 0.221 | 7 | 0.810 | 0.770 | 0.052 | 8 | 0.681 | 0.730 | **0.005** |
| Ilo16 | 169–205 | 3 | 0.522 | 0.634 | 0.350 | 7 | 0.353 | 0.680 | **0.000** | 5 | 0.276 | 0.606 | **0.000** | 6 | 0.647 | 0.742 | 0.583 | 7 | 0.571 | 0.565 | 0.464 |
| Ilo06 | 188–268 | 12 | 0.960 | 0.824 | **0.023** | 17 | 0.881 | 0.839 | **0.000** | 10 | 0.788 | 0.779 | 0.105 | 9 | 0.591 | 0.845 | **0.014** | 16 | 0.952 | 0.863 | 0.202 |
| Ilo23 | 242–294 | 8 | 0.455 | 0.773 | **0.000** | 8 | 0.692 | 0.717 | **0.014** | 7 | 0.636 | 0.719 | **0.004** | 4 | 0.652 | 0.691 | 0.213 | 12 | 0.702 | 0.809 | **0.000** |
| Ilo21 | 165–276 | 23 | 0.680 | 0.921 | **0.002** | 27 | 0.786 | 0.912 | **0.019** | 15 | 0.576 | 0.821 | **0.000** | 17 | 0.652 | 0.876 | **0.011** | 28 | 0.768 | 0.937 | **0.000** |
| Ilo11 | 234–274 | 7 | 0.360 | 0.784 | **0.000** | 6 | 0.447 | 0.809 | **0.000** | 6 | 0.394 | 0.723 | **0.000** | 4 | 0.318 | 0.706 | **0.000** | 8 | 0.441 | 0.757 | **0.000** |
| Ilo03 | 167–199 | 5 | 0.320 | 0.619 | **0.001** | 7 | 0.524 | 0.737 | **0.000** | 6 | 0.545 | 0.709 | **0.012** | 5 | 0.435 | 0.747 | **0.000** | 8 | 0.545 | 0.717 | **0.000** |
| Across loci | | 9.158 | 0.701 | 0.773 | **0.000** | 11.053 | 0.767 | 0.798 | **0.000** | 8.842 | 0.713 | 0.771 | **0.000** | 8.316 | 0.692 | 0.768 | **0.000** | 11.947 | 0.742 | 0.796 | **0.000** |

**Notes.**

Ra, allelic size range; Na, average number alleles per locus; $H_O$ and $H_E$, observed and expected heterozygosity, respectively; P, statistical significance for tests of departure of Hardy–Weinberg equilibrium.

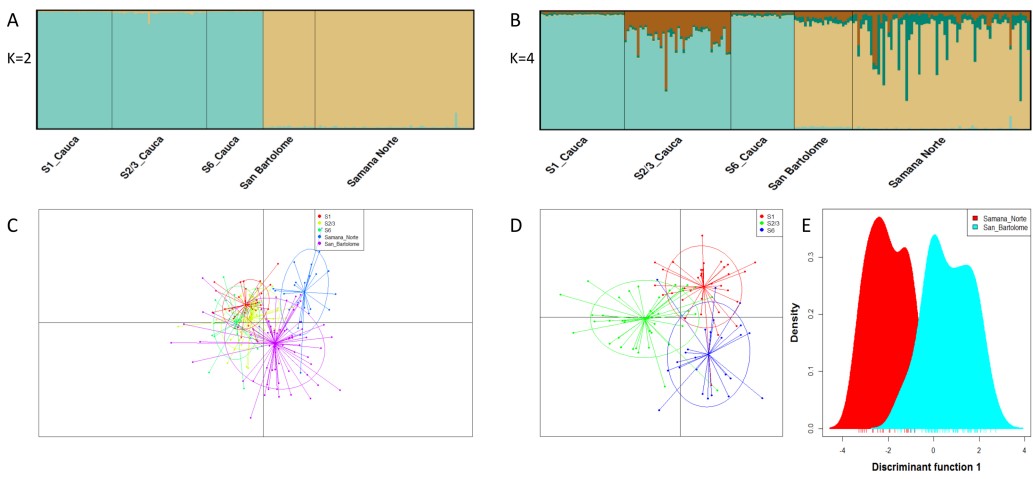

**Figure 2** **Population structure suggested by STRUCTURE and Discriminant analysis of principal components.** (A, B) Bar plots of population ancestry coefficients as estimated by STRUCTURE. Plots are provided for $K = 2$ and 4. The $q$-values were consensus estimates produced by CLUMPP across 20 iterations of STRUCTURE. In (C–E), Discriminant analysis of principal components including the full set of 19 microsatellite loci and three Colombian rivers (C), three sections of Cauca River (D) and the rivers San Bartolomé and Samaná Norte (E). This analysis utilized 38 Principal Component Analysis and the first two linear discriminants.

**Table 3** **Pair-wise Jost's *Dest* (upper diagonal) and *F'*st (below diagonal) among samples of *I. longirostris* from the Colombian rivers Cauca, San Bartolomé and Samaná Norte.**

|              | S1_Cauca | S2/3_Cauca | S6_Cauca | San Bartolomé | Samaná Norte |
|--------------|----------|------------|----------|---------------|--------------|
| S1_Cauca     |          | **0.019**  | **0.020** | **0.043**     | **0.076**    |
| S2/3_Cauca   | **0.010** |           | 0.001    | **0.039**     | **0.039**    |
| S6_Cauca     | **0.013** | 0.009     |          | **0.052**     | **0.039**    |
| San Bartolomé | **0.016** | **0.014** | **0.019** |              | **0.038**    |
| Samaná Norte | **0.016** | **0.010** | **0.012** | **0.013**     |              |

**Notes.**
Values in bold denote statistical significance.

not be an important component of commercial species' migrations. It remains to explore whether potential differences in spawning periods or reproductive/alimentary behavior with commercial species may explain the low captures.

The three approaches for measuring population genetic structure generated different, but non-excluding results. STRUCTURE revealed only two stocks related to the highest hierarchical grouping, which is concordant with other studies that show that this software is limited for the nested fine substructure detection (*Evanno, Regnaut & Goudet, 2005*) and for the cluster identification at low levels of genetic differentiation (*Latch et al., 2006*). Except for genetic difference between S2/3 and S6, the results of the discriminant analysis and AMOVA (fixation and genetic differentiation indexes) are similar and reveal a level of fine scale structuring, indicating that the suggested structure is reliable. These latter analyses support the idea that *I. longirostris* is structured in four (AMOVA) or five stocks (discriminant analysis).

Genetic differences between fishes from different sections of Cauca and Magdalena rivers were also found in populations of *P. magdalenae* (*Orozco Berdugo & Narváz Barandica, 2014*). This outcome might be explained by the short-distance migration range and habitat preferences of *I. longirostris* considering that the physicochemical characteristic of the habitat may explain the genetic structure in populations of other species (*Schaack & Chapman, 2003*; *Duponchelle et al., 2006*; *López-Macias et al., 2009*). Another alternative could be that homing behavior explains the genetic structure, which occurs in other members of Prochilodontidae (*Godoy, 1959*; *Godoy, 1975*; *Godinho & Kynard, 2006*).

Additionally, the minor genetic diversity upstream of the steep topography in the Cauca River may indicate that rapids do limit the gene flow between these sectors. However, the genetic differences between S6 and S2/3 in the absence of topographic accidents suggest that the behavior of the species, rather than the physical barriers, plays an important role in the non-homogeneous distribution of the genetic diversity. This explanation is also consistent with genetic differences found between Samaná Norte and San Bartolomé River.

In summary, this study developed the first set of polymorphic microsatellite loci for population genetics of *I. longirostris* and provides the first insights about the genetic structure of this species. Genetic differences were found among rivers and even within several sections of the Cauca River indicating that *I. longirostris* is conformed by, at least, four (likely five) stocks in the examined sites. This information, previous to the hydropower station construction, is crucial for monitoring the genetic diversity for management and conservation of this species as well as for complementing the genetic studies in Prochilodontidae.

## ACKNOWLEDGEMENTS

The authors thank to the Centro Nacional de Secuenciación Genómica, Universidad de Antioquia (Medellín, Colombia) for assistance in bioinformatics analysis.

### Funding

Universidad Nacional de Colombia and Integral S.A., on 19th September 2013 and Universidad Nacional de Colombia, Sede Medellín and Empresas Públicas de Medellín, Grant CT-2013-002443 "Variación genotípica y fenotípica de poblaciones de especies reófilas presentes en el área de influencia del proyecto hidroeléctrico Ituango." The funders had no role in study design, data collection and analysis, decision to publish, or preparation of the manuscript.

### Grant Disclosures

The following grant information was disclosed by the authors:
Universidad Nacional de Colombia and Integral S.A.
Universidad Nacional de Colombia, Sede Medellín and Empresas Públicas de Medellín: CT-2013-002443.

## Competing Interests

The authors declare there are no competing interests.

## Author Contributions

- Ricardo M. Landínez-García conceived and designed the experiments, performed the experiments, analyzed the data, contributed reagents/materials/analysis tools, wrote the paper, prepared figures and/or tables, reviewed drafts of the paper.
- Edna J. Márquez conceived and designed the experiments, analyzed the data, contributed reagents/materials/analysis tools, wrote the paper, prepared figures and/or tables, reviewed drafts of the paper.

## Animal Ethics

The following information was supplied relating to ethical approvals (i.e., approving body and any reference numbers):

The preserved tissues of this fishery resource were provided by Integral S.A. (Scientific cooperation agreement between Universidad Nacional de Colombia and Integral S.A., on 19th September 2013).

## Data Availability

The raw data has been supplied as Supplemental Dataset.

## Supplemental Information

Supplemental information for this article can be found online at http://dx.doi.org/10.7717/peerj.2419#supplemental-information.

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
