# Peer review of "Development and characterization of 24 polymorphic microsatellite loci for the freshwater fish Ichthyoelephas longirostris (Characiformes: Prochilodontidae)"

_PeerJ, doi:10.7717/peerj.2419_

## Round 0.1 · original submission · Minor Revisions

· Academic Editor

Minor Revisions

I am sorry for the delay in getting this back to you - we were waiting on a third referees who just backed out on compleiing the review, but the other two referees have detailed and similar reviews, so I am happy to move forward with these. There are two primary issues that must be dealt with prior to the paper being acceptable for publication.

The first is that the journal has decided that microsatellite primer notes do not fit the criteria of the Research Articles category, and have adopted a policy that there must be some sort of biological analysis to complete the characterization of the microsats rather than simply a primer note. You will need to provide some biological analysis using these primers to complete the work prior to it being acceptable for publication.

The second issue.is the language use and grammar. Both referees have provided extensive comments to improve the English of the article (please check for the marked-up manuscript attachments), but a quick read indicates that they have missed some also. I would suggest that you find a native English speaker to assist you with your edits if at all possible, and can suggest some options for you if you need assistance.

I look forward to seeing your revised manuscript.

·

Basic reporting

The manuscript describes the isolation and characterization of a set of 24 polymorphic microsatellites obtained by next-gen sequencing technology in a freshwater fish species from Colombia. In general, the manuscript is adequate to the journal scope as PeerJ has been published similar papers, typically characterized by a writing structure of a primer note description.
The results presented in this manuscript are very useful to any one who wants to study population genetics of Ichthyoelephas longirostris, a very interesting species of Prochilodontidae, appearing in a basal branch of the family phylogeny. I would suggest include at least one or two sentences in the Introduction section presenting such importance of this species, besides its conservation concerning.
The comparison of levels of observed heterozygosity between species (Discussion) should be rewrite. Why would it be expected that such levels should be “concordant”?
Moreover, there are several grammatical mistakes and an extensive revision by an English native is imperative.
Minor points are directly indicated in the pdf attached. I would suggest rewrite the Abstract without mention to the equipment brand in which the large-scale sequencing was made; and the same in the keywords.

Experimental design

No additional comments. The authors used proper methods and analyses to present their list of isolated microsatellites for the species studied.

Validity of the findings

No additional comments.
The results presented in this manuscript are very useful to any one who wants to study population genetics of Ichthyoelephas longirostris, a very interesting species of Prochilodontidae, appearing in a basal branch of the family phylogeny. I would suggest include at least one or two sentences in the Introduction section presenting such importance of this species, besides its conservation concerning.
The comparison of levels of observed heterozygosity between species (Discussion) should be rewrite. Why would it be expected that such levels should be “concordant”?
Minor points are directly indicated in the pdf attached. I would suggest rewrite the Abstract without mention to the equipment brand in which the large-scale sequencing was made; and the same in the keywords.

Additional comments

No additional comments.

Reviewer 2 ·

Basic reporting

This article describes a new set of microsatellite markers for an endemic freshwater fish endemic to a river basin in Colombia. These markers will be an important tool to investigate the evolution and ecology of a species belonging to a very interesting assemblage of fishes for which very little is known.

The manuscript meets the standards of such a publication. However, it is necessary to have the article checked by an English speaker prior to publication.

Adding one or two sentences in the introduction describing the basic ecology of the species would add value to the article (e.g. say it is a freshwater fish).

I added a few minor comments to the PDF document attached.

Experimental design

the experimental design is appropriate

Validity of the findings

The findings are valid

Additional comments

No additional general comments

Annotated reviews are not available for download in order to protect the identity of reviewers who chose to remain anonymous.

---

## Round 0.2 · Minor Revisions

· Academic Editor

Minor Revisions

I have reviewed your resubmission, and you have done a good job of addressing the concern of the referees that the initial submission did not address a biological question, but the new sections need some additional work before the paper can be accepted. There are issues with grammar and language usage that need to be resolved before this manuscript can be accepted. I have sent you a word document with tracked changes suggesting edits throughout and am willing to work with you directly to finalize the edits prior to final acceptance. Please look for my email with the marked-up manuscript.

---

## Round 0.3 · accepted · Accept

· Academic Editor

Accept

Thank you for your detailed edits of the manuscript. You have addressed all the concerns that the referees and I had with your initial submission, and I am happy to accept the revised manuscript and move it forward into production at this time.